# Hierarchical zwitterionic modification of a SERS substrate enables real-time drug monitoring in blood plasma

Fang Sun[1,*], Hsiang-Chieh Hung[1,*], Andrew Sinclair[1], Peng Zhang[1], Tao Bai[1], Daniel David Galvan[1], Priyesh Jain[1], Bowen Li[2], Shaoyi Jiang[1,2] & Qiuming Yu[1]

Surface-enhanced Raman spectroscopy (SERS) is an ultrasensitive analytical technique with molecular specificity, making it an ideal candidate for therapeutic drug monitoring (TDM). However, in critical diagnostic media including blood, nonspecific protein adsorption coupled with weak surface affinities and small Raman activities of many analytes hinder the TDM application of SERS. Here we report a hierarchical surface modification strategy, first by coating a gold surface with a self-assembled monolayer (SAM) designed to attract or probe for analytes and then by grafting a non-fouling zwitterionic polymer brush layer to effectively repel protein fouling. We demonstrate how this modification can enable TDM applications by quantitatively and dynamically measuring the concentrations of several analytes—including an anticancer drug (doxorubicin), several TDM-requiring antidepressant and anti-seizure drugs, fructose and blood pH—in undiluted plasma. This hierarchical surface chemistry is widely applicable to many analytes and provides a generalized platform for SERS-based biosensing in complex real-world media.

[1] Department of Chemical Engineering, University of Washington, Box 351750, Seattle, Washington 98195, USA. [2] Department of Bioengineering, University of Washington, Seattle, Washington 98195, USA. * These authors contributed equally to this work. Correspondence and requests for materials should be addressed to S.J. (email: sjiang@uw.edu) or to Q.Y. (email: qyu@uw.edu).

Blood plasma and serum hold the most valuable biochemical information for clinical diagnostics, but remain notoriously difficult to analyse without extensive processing. Rapid or real-time analyte detection in blood is particularly critical to therapeutic drug monitoring (TDM), which quantitatively measures the blood concentration of medications with a narrow therapeutic range[1]. TDM is currently a logistically complex and expensive process, and techniques to accurately monitor plasma drug concentrations in real time could dramatically simplify TDM and expand its reach.

Surface-enhanced Raman scattering (SERS) is one of the most sensitive spectroscopic techniques available and this ultrasensitivity combined with its label-free molecular specificity promise to make SERS a prominent factor in next-generation diagnostics[2–4]. SERS can be adapted to a wide range of detection targets, from small organic biomolecules and drugs to proteins, nucleic acids, cells and microorganisms[5–7]. Therapeutic drugs are typically excellent candidates for SERS detection, as ~95% of marketed drugs contain a conjugated ring system (such as a benzene ring)[8], which tend to produce the relatively large Raman scattering cross-sections necessary for high sensitivity. To date, several drugs have been directly identified in saliva and urine using SERS, but blood samples require separation and chromatographic purification before SERS detection[9–12]. The Raman-scattering enhancement seen in SERS decreases sharply when analytes are too far from a SERS-active surface[13,14]. In blood, the wide assortment of small molecules (for example, metabolites, carbohydrates, lipids and nucleotides) and plasma proteins compete with target analytes to bind the metallic SERS substrate[15,16]. This competing adsorption, known as fouling, blocks analytes from reaching SERS-active substrate 'hotspots' and generates substantial background noise, strongly reducing assay sensitivity and specificity. Analytes with weak affinity to SERS substrates or with small intrinsic Raman cross-sections present further difficulties. To solve the apparently contradictory challenges of resisting nonspecific fouling, while permitting or even promoting the diffusion of target analytes to SERS-active substrates, creative new surface chemistry modification approaches are necessary.

Here we present such an approach by functionalizing the SERS optofluidic system (shown in Fig. 1a) with a hierarchical zwitterionic modification. This modification contains two layers: a self-assembled monolayer (SAM) of 'attracting' or 'probing' functional thiols closest to the SERS-active substrate to physically attract analytes with weak surface affinity or chemically amplify the signals of analytes with small Raman activity and a second layer of non-fouling zwitterionic poly(carboxybetaine acrylamide) (pCBAA) grafted via surface-initiated atom transfer radical polymerization (SI-ATRP) to protect the 'hotspots' from the barrage of proteins in whole blood plasma that would typically limit detection sensitivity (Fig. 1b). We used this system to quantify the dynamic concentration of anticancer drug doxorubicin (DOX) in undiluted human blood plasma and demonstrated continuous real-time monitoring of the free DOX concentration with high sensitivity and accuracy alongside a rapid response time. The hierarchical modification also enabled detection of several TDM-requiring drugs, as well as blood fructose and pH. As this surface chemistry is widely applicable to many analytes, this strategy provides a generalized platform for real-world SERS-based biosensing directly and continuously in complex media.

## Results

### The necessity of zwitterionic modification.
Zwitterionic materials such as poly(carboxybetaine) have been used for a wide range of medical and engineering applications[17–20]. These superhydrophilic polymers demonstrate exceptionally low fouling and high long-term stability in complex physiological fluids. To demonstrate the necessity of zwitterionic modification on a SERS substrate encountering complex media, we selected rhodamine 6G (R6G) as a model analyte; R6G is a widely used dye with a large Raman cross-section[21]. In this and all other experiments presented here, we used whole, undiluted human blood plasma. Using a SERS optofluidic system, four solutions were pumped over bare gold and pCBAA-modified SERS substrates in succession: (1) saline buffer (PBS), (2) plasma, (3) PBS again and (4) plasma spiked with 1 μM R6G. Each solution was circulated for 10 min and SERS spectra were continuously acquired. On the bare gold surface, spectra remained constant after 8 min of plasma exposure and a representative spectrum is shown in Fig. 2a. The SERS spectrum of plasma is complex and displayed several characteristic peaks similar to those found in previous reports[15,22]; detailed assignments are summarized in Supplementary Table 1. Notably, the spectrum remained unchanged when PBS was subsequently delivered (step 3) in an attempt to rinse the substrate (Fig. 2a), suggesting that plasma proteins had fouled the bare gold surface and were difficult to remove. When we finally flowed R6G-spiked plasma over the substrate, the same spectrum still remained—despite the dye's large Raman scattering cross-section, the previous fouling completely blocked its signals (Fig. 2a). Comparatively, R6G at the same concentration exhibited extremely strong SERS signals on a clean, unfouled SERS substrate (Supplementary Fig. 1).

The blue spectrum shown in Fig. 2b is the baseline SERS signal of the pCBAA-modified surface in PBS. The spectrum is very similar to that of a 1-undecanethiol SAM, as most of the peaks come from the alkane-based ATRP initiators[23]. The signal from pCBAA itself is minimal, as pCBAA is an aliphatic polymer and is separated from the gold surface by ~2.2 nm. Peak assignments are listed in Supplementary Table 2. In contrast to the rapid and irreversible spectral changes seen when an unmodified gold surface was exposed to plasma, SERS signals from the pCBAA-modified surface remained constant when switching from PBS to plasma (red spectrum in Fig. 2b). This suggested that nonspecific adsorption was reduced to an undetectable level. Strong new signals appeared when we flowed R6G-spiked plasma over the surface (Fig. 2b, green). When baseline peaks were subtracted, the final spectrum (Fig. 2b, purple) clearly showed the characteristic R6G peaks.

The optofluidic system allowed us to monitor molecular adsorption onto the SERS-active surface in real time. We used the amide I vibrational band at 1,649 cm$^{-1}$, to track protein adsorption from the plasma[24], and the C–C stretching band at 1,508 cm$^{-1}$, to monitor R6G. Figure 2c shows the intensity change of these peaks over time from alternating buffer and plasma solutions. To further quantify protein adsorption on bare and pCBAA-modified surfaces, we used an surface plasmon resonance (SPR) biosensor. SPR sensorgrams revealed dense protein coverage on the bare surface (452 ng cm$^{-2}$), whereas only 0.3 ng cm$^{-2}$ was seen on the pCBAA-modified surface (Fig. 2d). The heavy protein fouling from plasma ruined the unmodified SERS sensor, whereas pCBAA modification effectively protected the SERS sensor.

Gold surfaces are often modified with oligo(ethylene glycol) (OEG)-based SAMs, which can effectively resist fouling from single protein solutions[25]. To compare the zwitterionic modification strategy reported here with OEG modification, we modified a SERS sensor with an OEG thiol (HS(CH$_2$)$_{11}$(OCH$_2$)$_4$OH) SAM and challenged it with human blood plasma. The OEG-modified sensor failed to detect the spiked R6G after flowing undiluted plasma (Supplementary

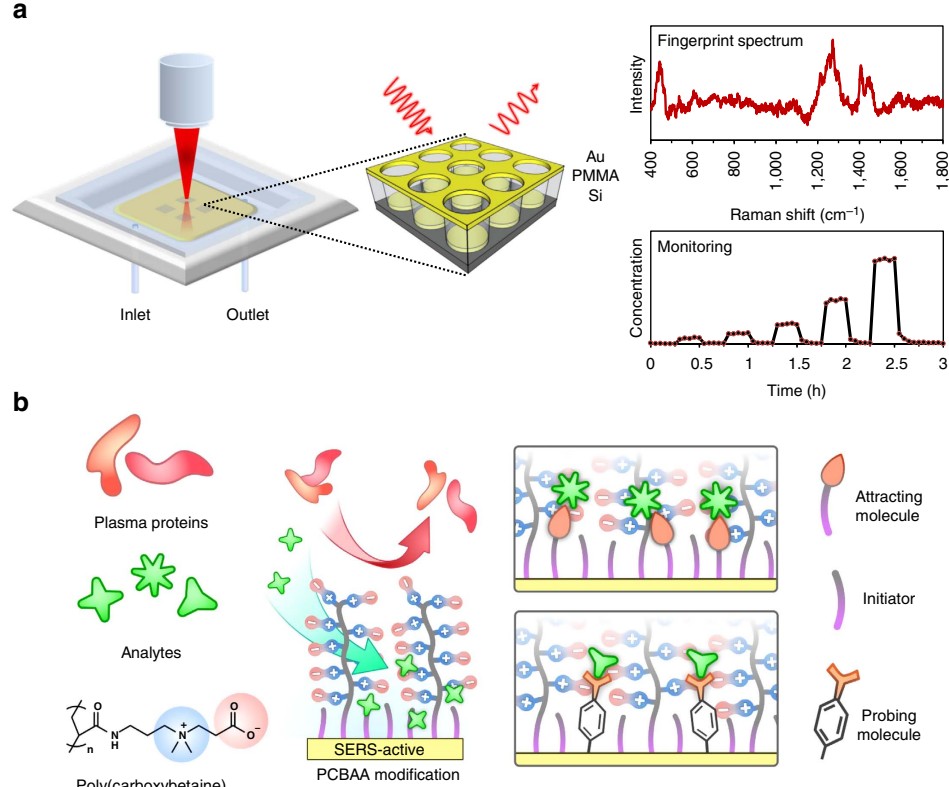

**Figure 1 | Schematic of Q3D-PNAs SERS optofluidic system and hierarchical zwitterionic surface modifications.** (**a**) Schematic of SERS optofluidic system incorporating a Q3D-PNAs SERS substrate to provide fingerprint spectra of analytes and quantitative, real-time monitoring. (**b**) Schematic of hierarchical pCBAA-based zwitterionic non-fouling modification on the SERS-active surface. Top: mixed SAM containing initiators and 'attracting' molecules, which have terminal functional groups that physically attract analytes to the surface for direct SERS detection. Bottom: mixed SAM containing initiators and 'probing' molecules (Raman reporters), which have functional groups that chemically interact with analytes to facilitate indirect analyte detection by monitoring changes in the SERS spectra of the probes.

Fig. 2a) and 111 ng cm$^{-2}$ of protein fouling was confirmed using an SPR sensor (Supplementary Fig. 2c). Both SERS and SPR results indicate that this short OEG SAM was insufficient to protect the SERS sensor from complex media. We conducted another comparison experiment by modifying the SERS substrate with a more hydrophobic SAM, 1-Undecanethiol (HS(CH$_2$)$_{10}$CH$_3$, C11 thiol). Again, no R6G could be detected after flowing plasma (Supplementary Fig. 2b) and even more protein fouling (263 ng cm$^{-2}$) was found on this C11 SAM-modified surface (Supplementary Fig. 2c).

**Real-time quantitative monitoring of DOX in plasma.** Continuous TDM can afford clinicians the opportunity to tailor therapeutic windows to individual patients, optimizing a drug's beneficial effects, while minimizing side effects. Ferguson *et al.*[26] developed a real-time drug tracking system based on unique aptamers that must be designed for each drug. In comparison, SERS-based biosensing is label free; if sensor fouling is mitigated, a SERS optofluidic system could be directly used for real-time TDM. We next aimed to use our pCBAA-modified SERS optofluidic platform to detect the popular chemotherapeutic agent DOX in plasma, as it exhibits significant pharmacokinetic variability[27]. DOX binds to plasma proteins in the bloodstream, rendering the protein-bound DOX inactive. When this DOX-containing plasma is analysed with a pCBAA-modified SERS sensor, we hypothesize that the polymer brush prevents these inactive DOX–protein complexes from reaching the SERS-active substrate—this enables measurement of only the active DOX concentration, which is of the most interest[28].

We added 20 µM DOX to plasma and flowed it through our pCBAA-modified detection system. The steady-state SERS spectrum was recorded (Fig. 3a, purple) and the baseline plasma spectrum (Fig. 3a, red) subtracted, to derive the pure DOX spectrum (Fig. 3a, blue), which matches those reported by other researchers[29]. Next, we recorded the SERS spectra of protein-free human plasma ultrafiltrate spiked with DOX over the clinically relevant concentration range of 0.05–10 µM; these subtracted DOX spectra are displayed in Fig. 3b. As expected, the intensities of characteristic DOX peaks increased with concentration. The DOX peak at 442 cm$^{-1}$ in the subtracted spectrum is attributed to C–C–O and C=O in-plane deformation. The 520 cm$^{-1}$ peak in the raw spectrum arises from Si in the SERS substrate and was used as an internal standard (shown in Supplementary Fig. 3). The relative magnitudes of peaks at 442 cm$^{-1}$ (in the subtracted spectra) to 520 cm$^{-1}$ (in the raw spectra) were selected to generate a detection curve illustrated in Fig. 3c, with a linear dynamic range between 0.05 and 2 µM shown in the inset.

Reversibility is an important characteristic for a viable sensor and determines the sensor response time to changes in analyte concentration. To test the reversibility of our pCBAA-modified SERS sensor, we injected DOX-spiked plasma (20 µM) at $t = 0$ and switched to pure plasma at $t = 250$ s. SERS spectra were continuously collected with a 30 s integration time and the relative intensity of $I_{442}/I_{520}$ over time was plotted (Fig. 3d). Clearly, DOX detection is reversible with exponential response constants of 43 and 95 s for partitioning and departitioning, respectively. With requisite parameters in hand, we conducted a model TDM experiment. SERS spectra were collected over 3 h as

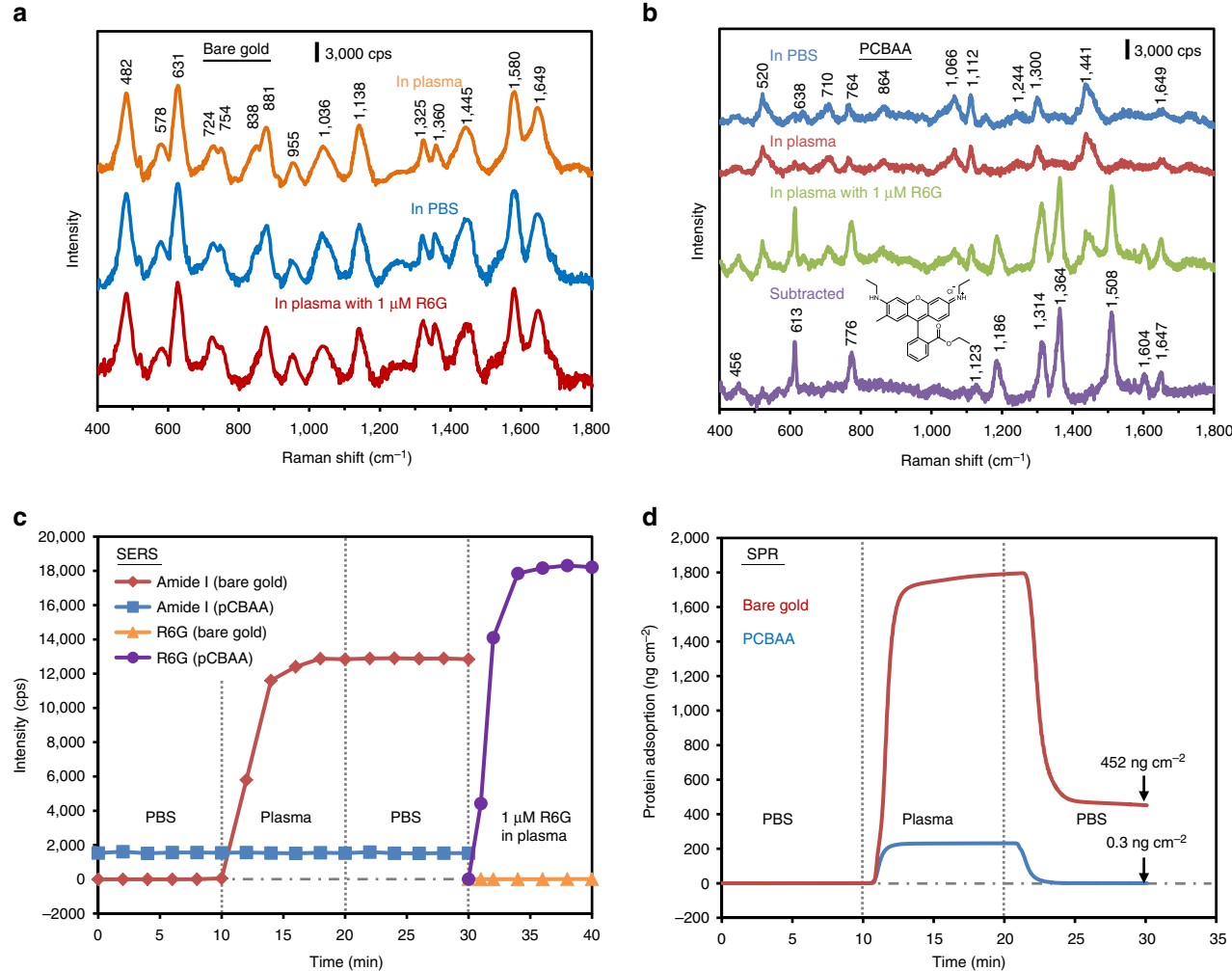

**Figure 2 | PCBAA surface modification enabling SERS detection in undiluted human blood plasma. (a)** SERS spectra acquired with the unmodified SERS optofluidic system. From top to bottom, spectra shown were recorded after flowing undiluted plasma, PBS and plasma spiked with 1 μM R6G each for 10 min. **(b)** SERS spectra acquired with the pCBAA-modified SERS optofluidic system. From top to bottom, spectra shown were recorded after flowing PBS, undiluted plasma and plasma containing 1 μM R6G each for 10 min, along with the subtracted spectrum from the last two. $\lambda_{ex} = 785$ nm, $P_{laser} = 1$ mW and $t = 30$ s with three accumulations. **(c)** Monitoring plasma protein adsorption and detecting R6G in plasma with the unmodified and pCBAA-modified SERS optofluidic system. Peak intensity of the amide I at 1,649 cm$^{-1}$ for protein and the C–C stretching at 1,508 cm$^{-1}$ for R6G were recorded as a function of time as PBS, plasma, PBS and R6G-spiked plasma were flowed sequentially. **(d)** Typical SPR sensorgram of protein adsorption from undiluted plasma on a bare gold and pCBAA-modified gold surface, showing protein adsorption of 452 and 0.3 ng cm$^{-2}$, respectively.

plasma flowed over the pCBAA-modified sensor; every 30 min, we alternated the inlet stream between pure and DOX-spiked plasma, the latter with DOX concentrations of 0.5, 1, 2, 4 and 8 μM. Based on the calibration curve generated in plasma ultrafiltrate, we plotted the DOX concentration detected in plasma (red points in Fig. 3e) compared with each spiked concentration. Of particular interest, all measured concentrations were less than spiked concentrations, indicating only free DOX was detected while protein–DOX complexes were blocked by the pCBAA brush as hypothesized. This is an important advantage of this platform and reinforces the importance of TDM to personalized dosing. Our results indicate that ∼23–33% of DOX molecules added to plasma remain unbound to proteins (Supplementary Fig. 4) and thereby pharmacologically active. To corroborate this drug–protein binding ratio, we also ultrafiltered the plasma samples immediately after SERS detection and quantified the free DOX with liquid chromatography–mass spectrometry (see Supplementary Methods). Using this method, 34–37% of DOX was calculated to be unbound (Supplementary Fig. 4) and the liquid chromatography–mass spectrometry

-quantified concentrations are also plotted in Fig. 3e (purple dashed line). Results from both analytical methods are similar to those reported in medical literature (∼29%)[30]. In addition, we found our system to be able to distinguish DOX from its metabolite doxorubicinol (DOXol), which shows a new characteristic peak at 1,348 cm$^{-1}$ (Fig. 3f). In these experiments, we flowed varying molar ratios of DOXol/DOX with a combined concentration of 2 μM and collected SERS spectra (Fig. 3f). The relative intensities of the 1,348 and 1,407 cm$^{-1}$ peaks illustrate a linear relationship as a function of DOXol/DOX ratio (Fig. 3g), indicating a multiplex detection capacity. Overall, we demonstrated the TDM capability of a pCBAA-modified SERS-active substrate when combined with a microfluidic system and successfully monitored the active level of DOX in plasma quantitatively and in real time.

**Incorporating 'attracting' thiols.** Both R6G and DOX were able to diffuse through the non-fouling pCBAA brush and penetrate or partition into the initiator SAM[31,32]. However, some analytes

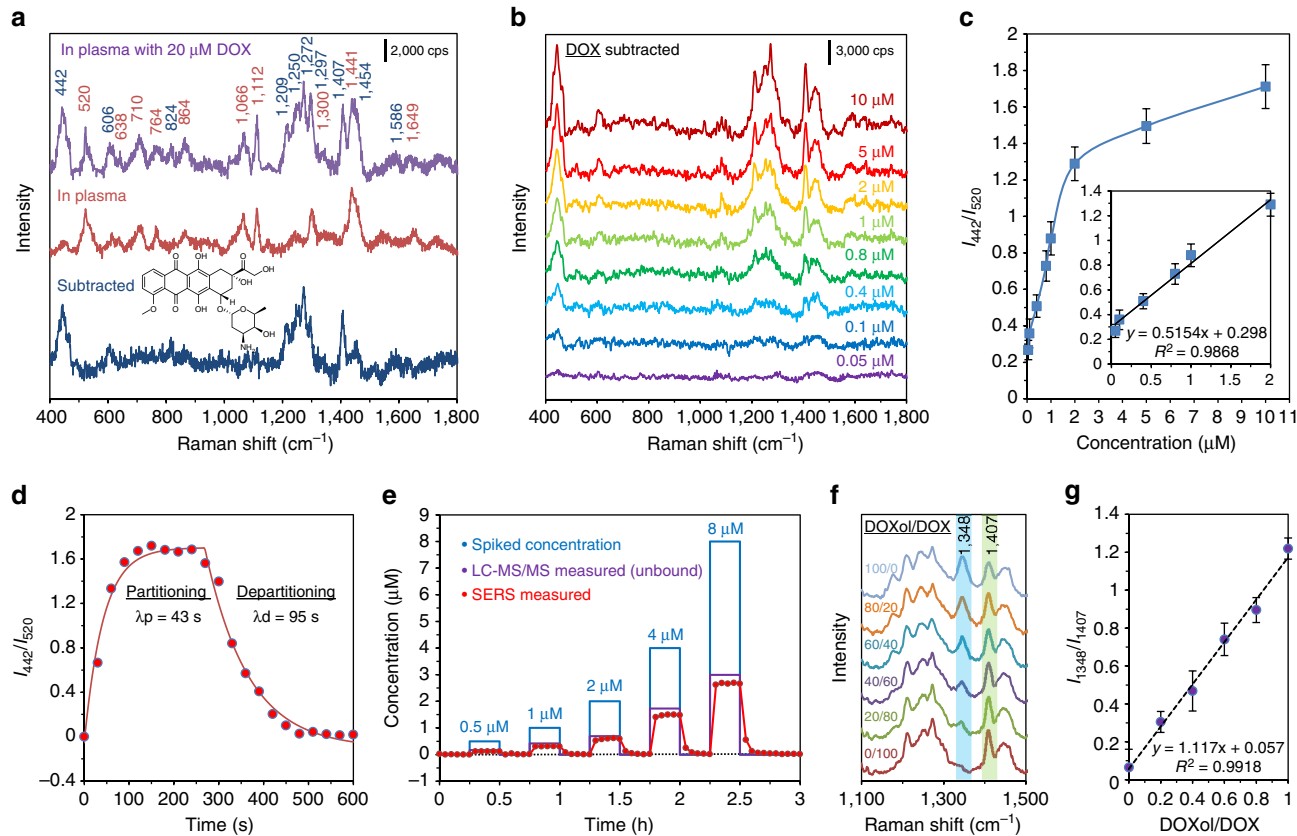

**Figure 3 | Real-time quantitative monitoring of DOX in undiluted human plasma using the pCBAA-modified SERS optofluidic system.** (**a**) SERS spectra of undiluted plasma spiked with 20 μM DOX, plasma and the subtracted spectrum. $\lambda_{ex} = 785$ nm, $P_{laser} = 1$ mW and $t = 30$ s with three accumulations. (**b**) Subtracted SERS spectra of DOX in plasma ultrafiltrate (UF) control at concentrations ranging from 0.05 to 10 μM. (**c**) DOX detection curve generated by plotting the average intensity ratio of the peaks at 442 and 520 cm$^{-1}$ as a function of the DOX concentration. Inset: a linear relationship was found for the physiologically relevant concentrations. The error bar stands for s.d. of three replicates. (**d**) Partitioning and departitioning of 20 μM DOX in plasma on a pCBAA-modified SERS-active surface. $\lambda_{ex} = 785$ nm, $P_{laser} = 1$ mW and $t = 10$ s with one accumulation. The $1/e$ time constants were calculated to be 43 s for partitioning and 95 s for departitioning. (**e**) *In vitro* real-time monitoring of DOX in undiluted human plasma (red) relative to spiked concentrations (blue) over the course of 3 h. The free DOX concentrations in plasma (red dots) were determined from the calibration curve in **c**. The free DOX concentrations in ultrafiltrated plasma (purple dashed line) were measured using liquid chromatography–mass spectrometry. (**f**) SERS spectra of pure DOXol and DOX, and DOXol/DOX mixtures. (**g**) Average intensity ratios of the peaks at 1,348 and 1,407 cm$^{-1}$ as a function of the DOXol/DOX ratios. The error bar stands for s.d. of three replicates.

are unable to partition into this SAM due to polar functional groups or low surface affinity and thus are undetectable by a system simply modified with a pCBAA brush. We selected three drugs that match this description and are prime candidates for TDM—tricyclic antidepressant amitriptyline hydrochloride (AH) and anti-seizure medications carbamazepine (CARB) and phenytoin (PHEN)[33,34].

To attract and concentrate these analytes closer to the SERS-active surface, we introduced shorter thiols with a functional terminal group mixed with the initiator forming the first layer. PCBAA brushes were then grafted from the initiator to form the second non-fouling layer. We selected 3-mercaptopropionic acid (3MA) and 1-propanethiol (C3) to form mixed SAMs with the initiator, to enable electrostatic and hydrophobic interactions with drug molecules, respectively. An optimized ratio of attracting (3MA or C3) to initiator thiols in the SAM was critical to maximize sensitivity, while maintaining a sufficient initiator (and thus pCBAA) density to strongly repel proteins. We generated a library of SAM-coated gold chips with varied ratios, targeting 0, 5, 10, 20, 40 or 100% of 3MA or C3 with the balance initiators. Then, we polymerized pCBAA brushes from each chip and evaluated fouling from undiluted plasma using an SPR sensor. Excellent non-fouling behaviour ($< 5$ ng cm$^{-2}$ adsorbed

proteins) was maintained when up to 20% of 3MA or 10% of C3 attracting thiols were incorporated (Supplementary Fig. 5a,c). The SERS spectrum resulting from the optimized mixed 3MA SAM (after pCBAA polymerization) is shown in Supplementary Fig. 5b (purple). As expected, it contains peaks similar to those observed from each modification alone (also shown in Supplementary Fig. 5b, with 3MA in red and initiator/pCBAA in blue). As 3MA has a larger Raman cross-section than the ATRP initiator, it produces stronger SERS peaks. By comparing the absolute intensities of the 3MA peaks in mixed and pure SAMs, we determined the final incorporated surface content of 3MA was 8%, with 92% initiator. We repeated this process for the optimized C3-containing mixed SAM and found the final C3 content to be 5% with the balance initiator; relevant SERS spectra are shown in Supplementary Fig. 5d. X-ray photoelectron spectroscopy and ellipsometry were further used to analyse the mixed SAMs (see details in Supplementary Methods) and both confirmed similar surface compositions as summarized in Supplementary Tables 3 and 4.

After optimizing the hierarchical mixed SAM chemistries incorporating 'attracting' molecules, we used these modified SERS substrates to analyse plasma spiked with 20 μM of one of the three selected drugs (AH, CARB or PHEN), to evaluate their

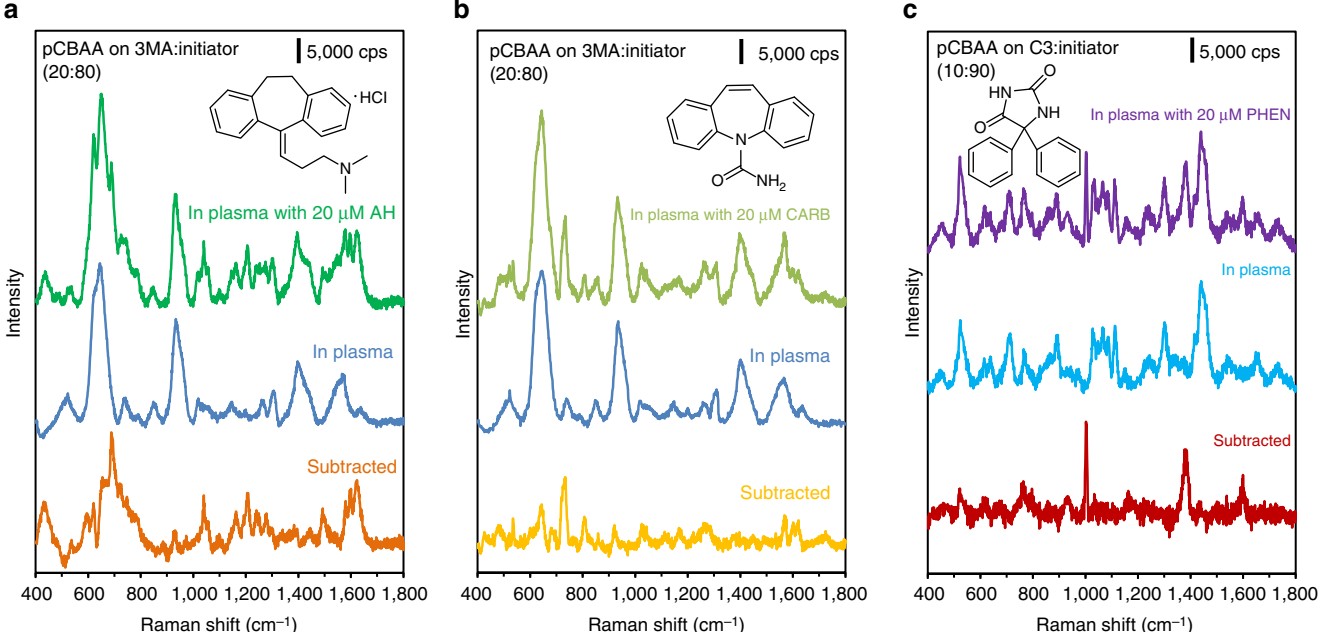

**Figure 4 | Hierarchical pCBAA surface modification to attract analytes and enable drug detection in undiluted human plasma.** (**a,b**) SERS spectra of 20 μM AH and 20 μM CARB spiked in undiluted plasma (green and olive), plasma (blue) and subtracted spectra (orange and yellow) acquired with the SERS optofluidic system hierarchically modified with pCBAA brushes grafted on the mixed SAM of 3MA and initiator (20 : 80). (**c**) SERS spectra of 20 μM PHEN spiked in undiluted plasma, plasma and subtracted spectrum acquired with the SERS optofluidic system hierarchically modified with pCBAA brushes grafted on the mixed SAM of C3 and initiator (10 : 90). $\lambda_{ex} = 785$ nm, $P_{laser} = 1$ mW and $t = 30$ s with three accumulations.

TDM potential. The hierarchical non-fouling surface containing 3MA (which is negatively charged) was applied to attract and detect positively charged AH and CARB, and the resulting SERS spectra are shown in Fig. 4a,b, respectively. Owing to the combined benefits of non-fouling pCBAA and electrostatic attraction from the mixed SAM, each of these drugs was clearly detected in undiluted plasma. Accordingly, Fig. 4c shows the SERS spectrum of PHEN detected with the hierarchical surface containing C3. The hydrophobic pockets created by C3 in the mixed SAM base layer may enhance PHEN detection by promoting surface partitioning. We also explored the limits of detection (LOD) achieved by our system for each of these three drugs in PBS. As shown in Supplementary Fig. 6a,b, the LOD was found to be 0.5 μM for CARB and 1 μM for PHEN, far below their clinically relevant ranges (20–50 μM for CARB and 40–80 μM for PHEN)[35]. The LOD for AH was 0.05 μM (Supplementary Fig. 6c), which is at the high end of its typical therapeutic window (0.04–0.05 μM)[35]. This suggests our SERS system could be useful for monitoring AH toxicity and overdose. In general, by mixing different 'attracting' molecules, this platform can be tailored to analytes with a broad range of physical and chemical properties.

**Incorporating 'probing' reporters.** Some analytes in blood have relatively small intrinsic Raman cross-sections, making them difficult to detect with SERS directly. These include mono-saccharides such as glucose and fructose, whereas protons or metal cations lack Raman activity completely[36,37]. To detect these analytes indirectly, probe molecules or Raman reporters have been developed. These probes are immobilized on SERS-active substrates to produce a specific response on interaction with corresponding analytes and they can even amplify detection signals. Detecting small fluctuations in blood pH is one important application of SERS probes, as these changes can influence how

drugs bind to plasma proteins and thus have an impact on their pharmacologically active concentration[38]. Likewise, the detection of monosaccharides in undiluted plasma is important for metabolic studies and diabetes treatment. Similar to our hierarchical chemistries incorporating 'attracting' thiols into the SAM, we designed mixed SAMs containing 'probing' thiols for these diagnostic applications.

We chose probe molecules 4-mercaptobenzoic acid (4MBA) and 4-mercaptophenylboronic acid (4MPBA) for these surface modifications to detect blood pH and fructose in plasma, respectively. As with the other hierarchical chemistries, we optimized the SAM ratios of each probe and initiators, and found that 5% of 4MBA or 4MPBA along with 95% initiator produced the best combination of signal strength and non-fouling after pCBAA was grafted. Figure 5a shows the SERS spectra observed in response to plasma pH changes (from 6.8 to 7.4), using the hierarchically modified sensor incorporating 4MBA. Owing to its extremely large Raman cross-section, 4MBA was responsible for all SERS signals, even though it comprised only 5% of the mixed SAM. All spectra were normalized to the peak at 1,080 cm$^{-1}$ (ν1 mode of benzene ring). The peak at 1,420 cm$^{-1}$, attributed to the symmetric COO$^{-1}$ stretching mode (νCOO$^{-1}$), increased concurrently with pH due to the deprotonation of 4MBA carboxyl groups[39]. By tracking the intensity change of the 1,420 cm$^{-1}$ peak, we were able to achieve pH detection in plasma with 0.1 pH unit resolution. Diminished pH sensitivity has been previously reported for SERS sensors modified solely with 4MBA, even in simple protein solutions[40]—the non-fouling pCBAA hierarchical modification we demonstrate here comparatively achieves robust and reliable pH detection in undiluted plasma with the same Raman reporter.

In a similar manner, we detected fructose in plasma using a hierarchical modification incorporating 4MPBA. The boronic acid head groups in 4MPBA can specifically bind to fructose, which breaks the symmetry of the probe molecule and induces

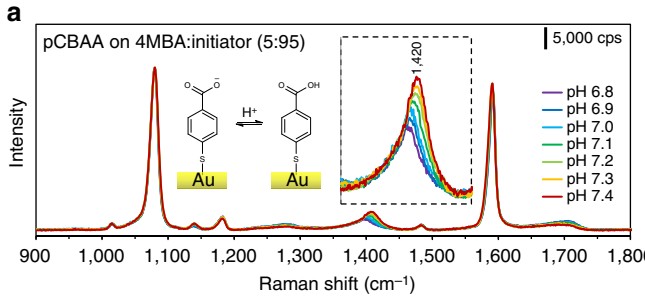

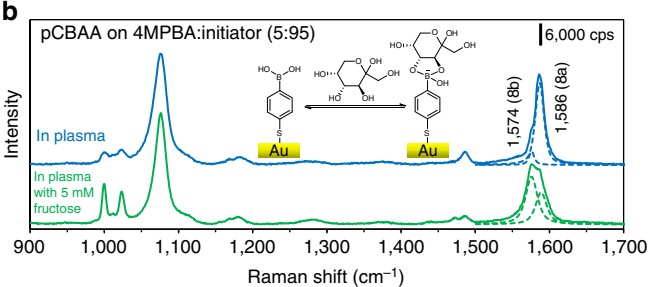

**Figure 5 | Hierarchical pCBAA surface modification with probing molecules to enable pH and fructose detection in undiluted human plasma.** (**a**) SERS spectra of 4MBA responding to pH-adjusted undiluted human plasma flowed over the hierarchically modified substrate with pCBAA grafted on the mixed SAM of 4MBA and initiator (5 : 95). (**b**) SERS spectra of 4MPBA responding to undiluted human plasma and plasma spiked with 5 mM fructose, flowed over the hierarchically modified substrate with pCBAA grafted on the mixed SAM of 4MPBA and initiator (5 : 95). $\lambda_{ex} = 785$ nm, $P_{laser} = 1$ mW and $t = 30$ s with a single accumulation.

ring reorientation and corresponding spectral changes. Our group has previously investigated fructose detection with 4MPBA-modified SERS sensors, in which we used mixed SAMs containing 4MPBA and short zwitterionic thiols to monitor fructose in single protein solutions[41]. At present, we demonstrate that fructose detection directly in plasma is possible using a hierarchical surface chemistry—the zwitterionic pCBAA brush is key to this advance, as it provides the best available protection from protein fouling. Figure 5b clearly shows the relative intensity change of the peaks at 1,574 and 1,586 cm$^{-1}$ (attributed to the 8a and 8b modes of the 4MPBA benzene ring), as fructose binds to the boronic acid moiety and breaks the probe's symmetry. Plasma was supplemented with 5 mM fructose for this detection experiment.

## Discussion

In complex, real-world media such as undiluted blood plasma, bare gold SERS sensors cannot survive. Nonspecific protein fouling rapidly and irreversibly blocks 'hotspots' on SERS-active substrates, nullifying the label-free sensitivity. To overcome this problem, we have introduced a novel modification to SERS-active substrate surfaces: a zwitterionic pCBAA brush coating grafted via SI-ATRP and designed to resist blood protein adsorption and protect the SERS sensor. Using this pCBAA-modified SERS substrate and an integrated optofluidic system, we demonstrated rapid, reliable and continuous DOX monitoring in undiluted human plasma. We further tailored the surface chemistry of the first layer by forming mixed SAMs containing functional thiols alongside the initiators required for polymer grafting. The additional electrostatic or hydrophobic interactions provided by these functional thiols allowed the attraction and detection of a typical antidepressant and two anticonvulsant drugs in undiluted plasma. Similarly, we used mixed SAMs featuring Raman reporter

probes to further extend the detection applications of this non-fouling hierarchical platform. The incorporation of these probe molecules enables SERS detection of more challenging analytes in undiluted plasma—whether they have weak SERS activity such as fructose or even no Raman activity at all such as the protons defining blood pH. Altogether, these hierarchical chemistries unified by nonfouling pCBAA have the potential to establish SERS optofluidic systems for real-time TDM in undiluted blood. Although the complexity and the cost of SERS sensor fabrication and surface modification need to be considered, we believe more simple and cost-effective processes will be developed with advances in nanotechnology and polymer chemistry. This hierarchical modification approach allows the first-layer functionality to be tuned for the best selectivity, while preserving the second zwitterionic polymer brush layer to effectively circumvent protein adsorption from complex media. This strategy could be widely adopted for a variety of biosensing applications.

## Methods

**Materials.** R6G (99%), DOX hydrochloride (DOX, 98.0–102.0%), AH (pharmaceutical secondary standard), CARB (pharmaceutical secondary standard), PHEN (pharmaceutical secondary standard), 3MA ($\geq$99%), C3 (99%), 1-undecanethiol (C11, 98%), 4-MBA (99%), 4-MPBA (90%), D-(−)-fructose ($\geq$99%), PBS packet (pH 7.4 and ionic strength 150 mM), copper(I) bromide (99.999%), copper(II) bromide (99.999%), 1,1,4,7,10,10-hexamethyltriethylenetetramine (97%) were purchased from Sigma-Aldrich (St Louis, MO). The OEG-terminated thiol (HS(CH$_2$)$_{11}$(OCH$_2$)$_4$OH) was purchased from ProChimia (Poland). The DOXol hydrochloride (DOXol, $\geq$90%) was purchased from Toronto Research Chemicals (Canada). Ethanol (200 Proof) was purchased from Decon Laboratories (King of Prussia, PA). Pooled human plasma (in sodium heparin, mixed gender) was purchased from Biochemed Services (Winchester, VA). High-purity deionized (DI) water was obtained with a Millipore water purification system with a minimum resistivity of 18.2 M$\Omega$ cm.

**Fabrication of SERS substrates and integration of the optofluidic system.** Quasi-3D gold plasmonic nanostructure arrays (Q3D-PNAs) SERS substrates[42] with high sensitivity and reproducibility were used in this work. The SERS substrates are composed of physically separated gold thin films with subwavelength nanoholes on the top and gold nanodiscs at the bottom as illustrated in Fig. 1a. They were fabricated via electron beam lithography following the same procedure reported previously[37]. On a 4-inch wafer, 70 pieces of SERS substrates, each with 4 of 50 × 50 μm Q3D-PNAs, were made to ensure high reproducibility from pattern to pattern and substrate to substrate. Ellipsometry (J.A. Woollam, α-SE) was used to measure the thickness of poly(methyl methacrylate) and gold coatings. The dimensions of the Q3D-PNAs were characterized using scanning electron microscopy (FEI Sirion), which are shown in Supplementary Fig. 7). The substrate was attached to a custom-made Teflon flow cell (1.2 cm × 1.2 cm × 1.5 mm) sealed with a glass cover. A peristaltic pump was used to deliver liquid samples.

**Preparation of hierarchical pCBAA films on SERS substrates or SPR chips.** The SI-ATRP initiator, mercaptoundecyl bromoisobutyrate and CBAA monomer were synthesized as described previously[43]. SAMs of the ATRP initiator on ultraviolet-ozone cleaned SERS substrates and SPR chips were formed by soaking bare substrates or chips overnight in 1 mM initiator solution in pure ethanol. On removal, the chips were rinsed with ethanol, DI water, ethanol and then dried and placed in a custom glass tube reactor under nitrogen atmosphere. Copper(I) bromide (7.17 mg) and copper(II) bromide (2.79 mg) along with a stir bar were placed in separate small test tubes, and 182.64 mg of CBAA monomer and a stir bar were placed in a large test tube. All three test tubes were sealed with a rubber septum and deoxygenated by eight repetitions of a strong vacuum followed by nitrogen back fill. Afterwards, 4 and 2 ml of deoxygenated water was added to the CBAA monomer and copper catalyst tubes, respectively. While stirring, 30 μl of 1,1,4,7,10,10-hexamethyltriethylenetetramine was added to the copper catalyst solution and stirred for 30 min for ligand complexation. Then, 160 μl of the copper catalyst solution was transferred to the 4 ml CBAA monomer solution. After stirring for 5 min, the entire CBAA monomer solution was transferred to the SERS or SPR chips contained in a tube reactor, to initiate polymerization. Reaction time was 30 min and a ~50 nm film (measured by ellipsometry) was successfully grown on SERS and SPR chips. The modified chips were rinsed with DI water and stored in PBS before SERS or SPR testing. The mixed SAMs were formed on the gold surface of SERS and SPR chips by immersing the chips in different compositions of 1 mM mixed thiol stock solutions in ethanol.

**Measurement of protein adsorption using SPR sensor.** A four-channel SPR sensor was applied to measure the nonspecific protein adsorption from human

plasma on the surfaces of SPR chips modified with hierarchical pCBAA films. The temperature controller was set to $25 \pm 0.01\,°C$. Protein adsorption was measured by sequentially flowing PBS, 100% human plasma and PBS over the modified surface each for 10 min at $40\,\mu l\,min^{-1}$ flow rate by a peristaltic pump. The wavelength shift between the baselines before protein injection and after rinsing with PBS was used to quantify the total amount of protein adsorbed. A reference channel containing a PBS flow was used for each chip to correct for baseline drift. A 1 nm wavelength shift from the SPR at 750 nm represents a surface coverage of $17\,ng\,cm^{-2}$ adsorbed proteins.

**Detection of drugs and small molecules using SERS optofluidic system.**
Raman spectroscopy was carried out on a Renishaw In Via Raman spectroscope connected to a Leica DMLM upright microscope. A $\times 50$/numerical aperture $= 0.8$ objective was used to focus a 785 nm laser on Q3D-PNAs and to collect the $180°$ scattered light from the sample surface. The laser power was set at 1 mW. A spectral resolution of $1.1\,cm^{-1}$ was achieved and spectra ranging from 400 to $1,800\,cm^{-1}$ were collected with an exposure time of 10 or 30 s and 1 or 3 accumulations. The flow rate was $40\,\mu l\,min^{-1}$.

**Data availability.** Data supporting the findings of this study are available within the article and its Supplementary Information files.

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

## Acknowledgements
This work was supported in part by National Science Foundation (NSF) (CBET 1264470, S.J.) and NSF (CBET 1159609, Q.Y.). Raman, electron beam lithography and X-ray photoelectron spectroscopy experiments were performed at the Molecular Analysis Facility (MAF) and the University of Washington–Washington Nanofabrication Facility (U.W. WNF). The MAF and WNF are part of the National Nanotechnology Coordinated Infrastructure (NNCI) at U.W. supported by NSF (ECCS-1542101).

## Author contributions
F.S., H.-C.H. S.J. and Q.Y. conceived and designed the study. F.S., H.-C.H., S.J., Q.Y., A.S., P.Z., T.B. and B.L. participated extensively in the scientific discussion about the study. F.S. and H.-C.H. performed surface modification, SERS and characterization experiments. B.L. conducted the free drug ratio measurement experiments. All authors contributed to preparing the manuscript.

## Additional information

**Competing financial interests:** The authors declare no competing financial interests.

**Publisher's note**: 

