## [Peer Review File · Nature Communications]

Reviewer #1 (Remarks to the Author):

In this work, the authors develop a surface-enhanced Raman spectroscopy (SERS) substrate co-modified with a zwitterionic species (intending to counteract biofouling) and another small molecule (intending to "attract" analytes of interest). Their aim was to specifically detect therapeutic drugs present in blood plasma. While the spectral results look promising, some critical characterization of the molecular character and plasmonic character of the substrate is missing. The use and interpretation of the achieved Raman spectra is naïve/overstated. In addition, I do not think this work is sufficiently novel to warrant publication in Nature Communications. There are a fair number of previous publications related to extrinsic SERS sensing in blood plasma using mixed monolayers (though somewhat different mixed monolayers than those used here). Some of these papers probe the same targets (pH, fructose) probed herein. Other significant concerns are detailed below:

1. In general, there's insufficient characterization of the plasmonic character of the sensor surface. The authors make claims about their betaine-based layer protecting "hot spots" but this claim is meaningless without some characterization of substrate enhancement factor before/after modification with surface modifications.
2. There's also insufficient characterization of the molecular character of the modified surfaces. The authors state what ratio of modifying molecules are presented to the surface but never demonstrate the adsorbed populations, even though the identity of the adsorbed SAMs are critical to the conclusions they make about sensing mechanism. The authors attempt to make a conclusion about coverage ratio among the two molecules presented using naïve logic. They claim that by "comparing the absolute intensities of the 3MA peaks in mixed and pure SAMs," they determined the percent of the surface covered with each of the presented molecules. This is not possible as they describe because it assumes that both molecules on the surface have the same Raman scattering cross-section (which they don't) and the same conformation (which they probably do not). The authors should employ traditional surface characterization techniques to achieve a clear understanding of surface chemistry since it is critical to their conclusions.
3. There is insufficient characterization of the surface-bound polymer. What is the molecular weight of the produced species? What is the PDI? Again, all this information is critical for anyone to understand the sensing mechanism or build on/repeat this work.
4. In addition, they should do surface characterization AFTER exposure to the target molecules and rinsing to demonstrate that the molecule of interest is present (perhaps with a technique such as SAMDI MS).
5. The authors should also include more specific molecular information about how they think (or what they know) about how the target analytes interact with the zwitterionic layer. Are species excluded based on size, charge, hydrophobicity? In general, I find the authors' language about the molecules buried deep in the poly-betaine SAM "attracting" the target molecules - the molecules flowing by in the plasma above the zwitterionic layer do not "know" that there's a hydrophobic interaction partner or a boronic acid buried nanometers away.
6. From an analytical perspective, the authors must present information about how many substrates were made/measured, the precision across substrate and between substrates, the stability of their assembled monolayers in the relevant matrices and across time. SERS substrates are notorious for inconsistent performance. There are also critical control experiments missing. The authors make

stark claims about the particularly effective performance of their betaine-based layer, but they only compare it to bare Au. They should really compare it to a non-zwitterionic SAM of similar length to make this claim. They also do not characterize the specificity of their sensor. Do they see changes in the Raman signal for the fructose sensor when other sugars are presented (likely because boronic acids are generally not sugar-specific)?

7. The authors also claim that they can use this sensor to detect what percentage of the DOX in plasma is free versus complexed with proteins (and made inactive). They claim that the ratio they calculate could be used in personalized drug dosing. This is a vast overstatement based on the data presented. The authors must measure the amount of DOX in the supernatant (supposedly the protein-complexed) and do a mass balance with the DOX they measure at the SERS substrate. When they make this claim, specifically that "36% of DOX molecules added to the plasma remain unbound to proteins and thereby physiologically active," they must benchmark this against the vast medical literature on this topic.

Reviewer #2 (Remarks to the Author):

The manuscript by Sun et al. reports a method to detect analytes with SERS in complex biological media (e.g. plasma, and possibly serum), directly or indirectly, by using a mixed monolayer designed to prevent serum proteins from adsorbing to the SERS-active surface, and at the same time to attract or directly bind the target analyte close to metal, thus taking advantage of the signal enhancement.

The manuscript is clearly written, well-structured and it properly addresses most relevant literature already published on the subject.

Overall, data presented are technically sound. Materials and methods used have been described in sufficient detail to allow for others to reproduce the experiments reported.

As evinced by the scant literature on qualitative and quantitative SERS detection of analytes in serum and plasma, the chemical complexity of these biofluids represents a formidable challenge for SERS. The interaction between SERS-active nanostructured metallic surfaces and biofluids and the consequences of such interaction on SERS sensing of target molecules are still not completely understood. The authors seem to be well aware of these issues, and main claim of their manuscript is a new way of functionalizing SERS metal surfaces with a which makes possible to obtain spectra of analytes in complex biofluids.

Although the same group published in 2015 a paper in which they present a simpler but similar approach to quantify fructose in presence of serum proteins (Sun et al. ACS Nano, 2015, 9 (3), pp 2668-2676), the results reported in this manuscript are definitely unprecedented: by using their "generalized platform" based on a hierarchical surface functionalization, the authors show that a range of structurally different molecules can be directly and indirectly detected (and quantified) in undiluted plasma. Everyone with experience in SERS knows that this detection, especially in undiluted plasma or serum, is far from trivial. In that sense, the general method presented in this manuscript represents a considerable step forward for the application of SERS to real-life samples. Thus this manuscript is likely to be of considerable importance for all the SERS community, and perhaps to further extend the interest for SERS to a broader public (e.g. clinicians).

In spite of these unquestionable merits, however, this manuscript also presents some points of

major and minor concern which in my opinion should be considered.

MAJOR REMARKS

The authors provide ample and convincing evidence to support the fact that structurally different molecules can be detected in undiluted plasma by using their approach. However, the limits of applicability of their approach are not mentioned. I wonder how much this research is really translational, and if it does really have the potential to be transferred into clinics, for real-life plasma samples.

In particular, I do have concerns about the following points.

1. The concentrations detected for the different molecules are rather high, if compared with those expected for real-life applications. Spectra with recognizable doxorubicin (DOX) bands (in buffer solutions) are shown for concentrations down to a 0.1 μM (Fig. 3b), whereas plasma spiked with 20 μM DOX, yields a SERS spectrum (Fig. 3a) with well-recognizable bands. Assuming that the limit of quantification (which was not determined) will reasonably be higher than the last detectable concentration of the "free" drug, this method is likely not to be widely useful for Therapeutic Drug Monitoring (TDM) of DOX, since typical peak concentrations in plasma (of total DOX, i.e. HAS-bound + free) are between 6 and 50 ng/mL (approx. corresponding to 10 to 80 nM) of total drug (see Sottani et al. *J. Chromatogr. B* 2013, 915-916, 71-78; Ackland et al. *Clin. Pharmacol. Ther.* 1989, 45, 340-347), with some exceptional cases having a total DOX concentration up to 950 ng/mL (approx. 1 μM , see Piscitelli et al. *Clin. Pharmacol. Ther.* 1993, 53, 555-561). Same arguments hold true for the other drugs directly detected: amitriptyline (AH), carbamazepine (CARB) and phenytoin (PHEN). They were all detected at a 20 μM concentration. Fructose was indirectly detected at 5 mM, two order of magnitude more than the usual normal physiological concentration (30-50 μM , according to the human metabolome database, www.hmdb.ca).

Considering that in SERS the electromagnetic enhancement factor is rapidly decreasing with the distance from the metal surface (Camden et al. *Acc. Chem. Res.* 2008, 41, 1653-1661; Kennedy et al. *J. Phys. Chem. B* 1999, 103, 3640-3646), in the approach making use of self-assembled monolayers of thiols proposed by the authors, the ability of attracting small molecules in presence of complex biofluids while avoiding fouling, is gained at the expense of enhancement (i.e. distance from the metal), leading to a decrease of limit of detection, limit of quantification and sensitivity with respect to "bare" metal surfaces.

I wonder if the sensitivity and limit of quantification of the method proposed can be good enough to be useful for a real TDM. Or, perhaps, if the method presented is an important step, but nevertheless just a step, toward a SERS-based TDM. The authors are kindly invited to critically discuss this point, perhaps adding a brief paragraph in their manuscript.

2. The novelty of the method presented consists in a two-fold role of the hierarchical SAM: i) preventing the fouling of the SERS-active surface and ii) directly attracting (or probing indirectly) the analytes of interest close to the same surface. The authors present SAMs incorporating different "attracting thiols" for the direct SERS detection of different molecules (i.e. alkane-based ATRP initiators for DOX, 3-mercaptopropionic acid for AH and CARB, 1-propanethiol for PHEN). What should be done in case another drug is to be detected, which is structurally different from the ones investigated in this manuscript? I wonder how general is this method: is there a rationale for selecting such "attracting" elements or it is a matter of trial and error? In the latter case, the method

would lack general applicability. The authors are kindly invited to explicitly address this issue in the manuscript.

3. Closely related to the previous point is the "selectivity" of the method proposed. For instance, can doxorubicinol (the main metabolite of doxorubicin, also present in the blood) be detected as well (and distinguished from doxorubicin) with the same method used for doxorubicin? Can the indirect method used to detect fructose be used to distinguish other sugars as well? The authors are invited, if not to present results from new experiments, at least to comment about this aspect.

MINOR REMARKS

Besides these major point of concern, there are few minor points which deserve some attention.

1. Band assignments for SERS spectrum of plasma (Table S1 in Supporting Information) are partially incorrect. Assignment of most bands in SERS spectra of plasma and serum is still debated, and is made mostly on the basis of the Raman shifts as found in literature from Raman spectra. On the other hand, some bands were previously assigned on the basis of direct comparison with the SERS spectra of the metabolites (Bonifacio et al. *Anal. Bioanal. Chem.* 2014, 406, 2355-1365, ref.15 of the manuscript), and can be relied on: bands at 631, 881 and 1138 cm^{-1} are due to uric acid, and the band at 955 is due to hypoxanthine and/or adenine (whose SERS spectra can be very similar). Collagen is very unlikely to be found in plasma, so it should not be considered as a possible candidate for assignments. The authors are kindly invited to consider some changes in Table S1.

2. The approach proposed involves complex, time-consuming, custom or expensive instrumentation or procedures (organic synthesis of initiators and of CBAA; EBL; UV-Ozone cleaner; custom glass tube reactor under nitrogen; deoxygenation steps; strong vacuum; etc.). In my opinion, the ease of substrate preparation and functionalization, the related costs and the substrate's shelf-life should be considered, and possibly addressed in the manuscript, when describing a new SERS method. The authors are kindly invited to address these issues, perhaps adding a sentence or a short paragraph in the manuscript.

3. Error bars are shown in Fig. 3c, but I could not find the numbers of independent measurements for each concentration. The authors are kindly invited to add this information to the manuscript.

In view of both merits and points of concern expressed above, my opinion is that, provided the authors adequately address all the remarks, an amended version of this manuscript should be definitely considered for publication in *Nat. Comm.*

Sincerely,
Alois Bonifacio

Reviewer #3 (Remarks to the Author):

This manuscript describes the development of a surface chemistry protocol that allow the detection of small molecules from blood samples by SERS. The manuscript is interesting and the results are compelling. The originality, however, is not that high since there are several examples in the literature of SERS detection from whole blood using different strategies (Nanomedicine : nanotechnology, biology, and medicine Volume: 12 Issue: 3 Pages: 633-41; ANALYST Volume: 141 Issue: 7 Pages: 2165-2174; ELECTROPHORESIS Volume: 37 Issue: 5-6 Pages: 786-789, CHEMICAL SCIENCE Volume: 6 Issue: 7 Pages: 4247-4254 Published: 2015). The use of zwitterionic surfaces to decrease surface fouling is also not new (JOURNAL OF MEMBRANE SCIENCE Volume: 475 Pages: 469-479 ; ADVANCED MATERIALS INTERFACES Volume: 3 Issue: 6 Pages: 646-646). In any case, the level of the science presented here is good and the work might be considered for publication after the minor points below are considered:

- 1) The use of the 520 cm^{-1} as an internal standard was a good idea. However, the band appears sometimes too weak in the spectra (see, for instance, figure 3). The concern is that division by a small (and noisy) number can lead to more errors. However, no errors are reported in the plots. Some statistical consideration should be added. It is actually not very clear that the 520 cm^{-1} is constant from the data in figure 3.
- 2) Although the experiments with single analytes were interesting, it is likely that blood samples from patients might contain mixtures of small molecules. How selective would be the layers for mixtures?

NCOMMS-16-05027

Response:

Reviewer #1 (Remarks to the Author):

In this work, the authors develop a surface-enhanced Raman spectroscopy (SERS) substrate co-modified with a zwitterionic species (intending to counteract biofouling) and another small molecule (intending to "attract" analytes of interest). Their aim was to specifically detect therapeutic drugs present in blood plasma. While the spectral results look promising, some critical characterization of the molecular character and plasmonic character of the substrate is missing. The use and interpretation of the achieved Raman spectra is naïve/overstated. In addition, I do not think this work is sufficiently novel to warrant publication in Nature Communications. There are a fair number of previous publications related to extrinsic SERS sensing in blood plasma using mixed monolayers (though somewhat different mixed monolayers than those used here). Some of these papers probe the same targets (pH, fructose) probed herein.

We appreciate your detailed comments and suggestions. Based on your input, we conducted additional experiments to better characterize the molecular and plasmonic character of the substrate and have revised the manuscript accordingly. All the details can be found in the following point-by-point responses.

It is substantially more challenging to effectively resist non-specific protein binding from complex, real-world media like undiluted blood plasma and serum than from simple media such as single protein solutions; this frequently results in biosensor failure in complex media. While SERS is one of the most sensitive sensing techniques, it is also highly vulnerable to this fouling process. Typical short self-assembled monolayers (SAMs) are often used to reduce fouling, but they cannot resist non-specific adsorption from 100% blood plasma or serum, and thus fail to protect the SERS surface under these harsh conditions. In the revised manuscript, we included results from new experiments we conducted to clarify this point. In one such experiment, we functionalized a SERS substrate surface with an oligo(ethylene glycol) (OEG) SAM, which has been widely used as an anti-fouling modification on gold surfaces and can effectively resist adsorption from single protein solutions. However, it failed to protect the SERS substrate from fouling when we tested it in undiluted blood plasma, with this fouling blocking our subsequent attempted detection of R6G. We confirmed this with SPR measurements.

As you mentioned, there are several previous publications demonstrating SERS detection in processed blood plasma. The samples in these studies were typically diluted, ultra-filtered, or purified to reduce or eliminate proteins—these processing steps would bottleneck future clinical TDM applications. Our current study demonstrates a new surface functionalization strategy that enables a SERS-based optofluidic system to continuously monitor drug concentrations in undiluted plasma. After using this system to successfully quantify several analytes with very different chemical and physical properties (e.g., Raman activities) in undiluted plasma, we believe this strategy provides a general solution to SERS biosensing in complex media, which has never been reported.

The pH and fructose detection experiments included here are mainly to demonstrate the concept that “probe” molecules can be incorporated into this platform to amplify the SERS signals of analytes with small Raman activities. This idea can also be applied to other analytes.

Other significant concerns are detailed below:

1. In general, there's insufficient characterization of the plasmonic character of the sensor surface. The authors make claims about their betaine-based layer protecting "hot spots" but this claim is meaningless without some characterization of substrate enhancement factor before/after modification with surface modifications.

Our group has extensively investigated the plasmonic properties of quasi-3D plasmonic nanostructure arrays (Q3D-PNAs), including the localized surface plasmon resonances (LSPR), the surface maximum local electric field intensity ($E_{\max}(\lambda_{\text{LSPR}})$), and the electric field distribution (which can clearly show the location of hot spots) (*Small* 7.3 (2011): 371-376; *Optics express* 19.21 (2011): 20493-20505). The electromagnetic enhancement factor (EF_{EM}) has been estimated from the relationship $(E_{\max}(\lambda_{\text{LSPR}})/E_0(\lambda_{\text{LSPR}}))^4$, where $E_{\max}(\lambda_{\text{LSPR}})$ and $E_0(\lambda_{\text{LSPR}})$ are the maximum local electric field intensity at the Au/air interface and the electric field intensity of incident light at the wavelength of LSPR, respectively. The EF_{EM} of the Q3D-PNA used in this work has been calculated to be over 1.2×10^8 (*Optics express* 19.21 (2011): 20493-20505).

Since the same Q3D-PNAs were used before and after surface modification, the EF_{EM} values would be equivalent. However, the SERS spectral intensity of R6G on an unmodified gold Q3D-PNA surface was about five times stronger than that on modified Q3D-PNA surfaces, as can be seen by comparing the SERS spectra shown in Fig. S1 and Fig. 2b. This difference could be due to three factors. First, the number of R6G molecules reaching modified and unmodified surfaces is likely different. The pCBAA-modified surface has a SAM composed of initiator molecules directly on the gold surface, along with the pCBAA polymer brushes. R6G molecules can readily diffuse through the polymer layer to reach the SAM, but the number of R6G molecules reaching the surface could be less than those reaching unmodified, bare gold surfaces. Second, the possible chemical enhancement due to the direct interaction of R6G with gold on the unmodified surface could be lacking on the pCBAA-modified surface. Third, since some of the R6G molecules may only be able to reach the top of the initiator SAM (~ 2.2 nm away from the gold surface), the local electric field intensity is weakened because of the rapidly decaying electric field.

Our intention here is not to compare enhancement before and after surface modification—rather, it's to point out that without pCBAA surface modification, fouling from whole blood plasma completely covers the Q3D-PNA surface, resulting in no SERS signal at all. For example, the SERS signal from target analytes in plasma on unmodified surfaces is undetectable—not due to a low enhancement factor, but purely because of the thick absorbed protein layer blocking access to the surface. The enhancement factor is not a meaningful

metric by which to evaluate SERS surface treatment efficacy in complex media like it can be in simple buffer solutions.

2. There's also insufficient characterization of the molecular character of the modified surfaces. The authors state what ratio of modifying molecules are presented to the surface but never demonstrate the adsorbed populations, even though the identity of the adsorbed SAMs are critical to the conclusions they make about sensing mechanism. The authors attempt to make a conclusion about coverage ratio among the two molecules presented using naïve logic. They claim that by "comparing the absolute intensities of the 3MA peaks in mixed and pure SAMs," they determined the percent of the surface covered with each of the presented molecules. This is not possible as they describe because it assumes that both molecules on the surface have the same Raman scattering cross-section (which they don't) and the same conformation (which they probably do not). The authors should employ traditional surface characterization techniques to achieve a clear understanding of surface chemistry since it is critical to their conclusions.

We took the reviewer's suggestion and further investigated the surface composition of mixed SAMs using both ellipsometry and XPS methods. Experimental details are provided in the Supporting Information, and the mixed SAM surface composition results found with ellipsometry and XPS are shown in Tables S3 and S4, respectively. The results from these two methods are comparable with those estimated from the SERS spectra.

As we mentioned in the manuscript, pure 3MA and C3 SAMs have much stronger SERS signals than the ATRP-initiator SAM, indicating that they have different Raman cross-sections. The absolute SERS intensity of the initiator SAM required enlargement by about 25 times to reach similar intensity to the pure 3MA SAM (red and blue lines in Fig. S5b). Please note that for the SERS spectrum taken from the hierarchical pCBAA-modified substrate (blue line), the SERS signal is only due to the initiator SAM because of the near-field effect. Similarly, the intensity of the initiator SAM signal had to be enlarged about 19 times to reach similar intensity as the pure C3 SAM (yellow and blue lines in Fig. S5d). Using the SERS spectra of pure and mixed SAMs to estimate the surface ratio, we assumed that the decreasing intensity of the 3MA peak at 935 cm^{-1} (or the C3 peak at 892 cm^{-1}) in the mixed SAM compared to that in the pure 3MA (or C3) SAM was due to the replacement of 3MA (or C3) molecules with initiator molecules on the surface. Thus we compared the relevant peak intensities (at 935 cm^{-1} for 3MA and 892 cm^{-1} for C3) of mixed and pure SAMs to obtain the surface ratios of mixed SAMs. In this way, we did not involve any peaks attributed to the initiator, and we did not assume 3MA (or C3) to have the same Raman cross-section as the initiator.

Because the surface ratio of initiator/3MA (or C3) influences the fouling property of hierarchical pCBAA-modified gold surfaces, a quick screening method we used was to form mixed SAMs with different initiator/3MA (or C3) ratios in stock solutions (as we displayed in Fig. S5a and c) and check the resulting protein adsorption with SPR. Overall, it is most crucial to determine the threshold ratio in bulk solution that results in non-fouling and leads to the intended SERS detection; knowing the final surface ratio would be a plus.

3. There is insufficient characterization of the surface-bound polymer. What is the molecular weight of the produced species? What is the PDI? Again, all this information is critical for anyone to understand the sensing mechanism or build on/repeat this work.

Over the past 11 years, our group has conducted extensive theoretical and experimental studies to understand the nonfouling mechanisms of zwitterionic polymers. (*Advanced Materials* 22.9 (2010): 920-932.) Compared to other polymerization methods, atom-transfer radical-polymerization (ATRP) is a highly controllable and repeatable reaction that results in a low PDI. In this work, we used surface initiated ATRP (SI-ATRP) to graft pCBAA polymers from the gold surface. Unlike polymers formed in solution, the molecular weight and PDI of surface-grafted polymers are typically difficult to directly measure; instead, the film thickness and refractive index are most often examined with ellipsometry to characterize these surface-bound polymers. Since we care about the nonfouling properties of grafted zwitterionic polymers, we also routinely evaluate protein adsorption from blood plasma and serum on modified surfaces using surface plasmon resonance (SPR) biosensors. To make it possible to repeat and understand this work, we have provided detailed experimental conditions and procedures for all the above in the manuscript and SI. We characterized the pCBAA films formed from the pure initiator SAM and mixed SAMs by measuring the film thickness and protein adsorption in plasma and serum with SPR, and all these measurements are provided in the manuscript and SI.

4. In addition, they should do surface characterization AFTER exposure to the target molecules and rinsing to demonstrate that the molecule of interest is present (perhaps with a technique such as SAMDI MS).

Since SERS is a very sensitive analytical technique and also has a near-field effect, we believe the characteristic analyte peaks in the SERS spectra were sufficient to demonstrate the presence of the analytes. Nonetheless, we took the reviewer's suggestion to further confirm the presence of analytes with mass spectroscopy. To do this, we exposed our SERS sensor to the analytes and then rinsed and collected the wash-out solution in PBS for analysis. The results are attached below, and confirm that R6G and DOX were detected in the wash-out solution at their exact molecular weights.

5. The authors should also include more specific molecular information about how they think (or what they know) about how the target analytes interact with the zwitterionic layer. Are species excluded based on size, charge, hydrophobicity? In general, I find the authors' language about the molecules buried deep in the poly-betaine SAM "attracting" the target molecules - the molecules flowing by in the plasma above the zwitterionic layer do not "know" that there's a hydrophobic interaction partner or a boronic acid buried nanometers away.

Indeed, the small analytes or target drugs do not "know" that there is an "attracting" layer when they are above the pCBAA polymer film. This film is dense enough to effectively repel protein adsorption, but leaves sufficient room for small analytes or drugs to easily diffuse or flow through. In addition, zwitterionic materials such as pCBAA strongly bind water molecules via electrostatically induced hydration, which both makes them resistant to protein interaction/fouling and have fewer interactions with other molecules (such as analytes and drugs). Through these mechanisms, the free-flowing analytes simply interact with the mixed SAM layer similar to how they would in the absence of a pCBAA brush, while the pCBAA simply screens out proteins and other macromolecules based primarily on size and strong hydration.

6. From an analytical perspective, the authors must present information about how many substrates were made/measured, the precision across substrate and between substrates, the stability of their assembled monolayers in the relevant matrices and across time. SERS substrates are notorious for inconsistent performance. There are also critical control experiments missing. The authors make stark claims about the particularly effective performance of their betaine-based layer, but they only compare it to bare Au. They should really compare it to a non-zwitterionic SAM of similar length to make this claim. They also do not characterize the specificity of their sensor. Do they see changes in the Raman signal for the fructose sensor when other sugars are presented (likely because boronic acids are generally not sugar-specific)?

As mentioned before, we designed the Q3D-PNA SERS substrates using FDTD simulations by investigating the plasmonic properties and the electric field distribution to ensure strong

enhancement and uniformly distributed electric fields. The Q3D-PNAs were fabricated using electron beam lithography (EBL) with a high-end EBL instrument that can correct the dose to ensure nanoholes are of a uniform size from the pattern corner to center. On a 4-inch silicon wafer, 70 chips and 4 Q3D-PNAs on each chip were made. Therefore, high reproducibility and uniformity were ensured from pattern to pattern and from chip to chip. Our previous works have also demonstrated the extraordinary reproducibility from pattern to pattern and batch to batch in the detection of bacteria, cells and small molecules (*Analytical Chemistry* 85.5 (2013): 2630-2637.; *Biosensors and Bioelectronics* 73 (2015): 202-207). We included the SEM images of the Q3D-PNA in the Supporting Information (Fig. S7). Regarding the stability of modified surfaces, while we did not test this on Q3D-PNA surfaces, we have tested monolayer stability on the similar flat gold surfaces of SPR chips; pCBAA-modified chips are stable for at least three months when stored in PBS buffer. Their non-fouling performance after storage is as good as when freshly made.

We took the reviewer's suggestion about additional control experiments. We formed SAMs of OEG thiol ($\text{HS}(\text{CH}_2)_{11}(\text{OCH}_2)_4\text{OH}$) and 1-Undecanethiol ($\text{HS}(\text{CH}_2)_{10}\text{CH}_3$, C11 thiol) on Q3D-PNA surfaces and followed the same procedure to conduct the SERS measurements by flowing PBS buffer, plasma and R6G-spiked plasma. After subtracting the SERS spectrum from R6G-spiked plasma from that of plasma, no detectable signal was seen in either control case, indicating protein fouling on both surfaces. The protein adsorption from undiluted blood plasma on the OEG SAM and C11 SAM were 111 ng/cm^2 and 263 ng/cm^2 , respectively, as measured with SPR. In contrast, protein adsorption on the pCBAA-modified surface was only about 0.3 ng/cm^2 . Discussion about these two control experiments were added to the manuscript on page 3, line 27. The experimental details as well as the SERS spectra and the protein adsorption results (Fig. S2) are provided in the Supporting Information.

The specificity of our SERS sensor has been demonstrated by the characteristic vibrational peaks of different molecules. Regarding the specificity of sugar molecules, we have addressed this issue in our previous work (*Analytical Chemistry* 86.5 (2014): 2387-2394). Briefly, the difference in binding affinity of different sugar molecules under normal pH was employed to realize detection specificity.

7. The authors also claim that they can use this sensor to detect what percentage of the DOX in plasma is free versus complexed with proteins (and made inactive). They claim that the ratio they calculate could be used in personalized drug dosing. This is a vast overstatement based on the data presented. The authors must measure the amount of DOX in the supernatant (supposedly the protein-complexed) and do a mass balance with the DOX they measure at the SERS substrate. When they make this claim, specifically that "36% of DOX molecules added to the plasma remain unbound to proteins and thereby physiologically active," they must benchmark this against the vast medical literature on this topic.

To confirm the free DOX concentration or unbound/bound drug ratio in blood plasma, we applied ultrafiltration to separate the free DOX and then used LC-MS to determine the concentrations and ratios. To further improve our experiments, we regenerated the calibration curve for SERS detection using DOX spiked in the control plasma ultra-filtrate rather than

simply PBS. We used the same batch of human blood plasma to prepare the samples for LC-MS and SERS experiments to exclude the variations induced by plasma samples. The results from ultrafiltration (about 38% free drug) are comparable to the results from SERS (about 31% free drug) and they are similar from other medical literature reported (about 29%, *Cancer Chemotherapy and Pharmacology* 38.6 (1996): 571-573.). Related discussion was added on page 4 of the manuscript, line 34, and experimental details were provided in the Supporting Information. Comparisons of the free DOX percentage obtained from the ultrafiltration method and SERS was provided in Fig. S4 in the Supporting Information.

Reviewer #2 (Remarks to the Author):

The manuscript by Sun et al. reports a method to detect analytes with SERS in complex biological media (e.g. plasma, and possibly serum), directly or indirectly, by using a mixed monolayer designed to prevent serum proteins from adsorbing to the SERS-active surface, and at the same time to attract or directly bind the target analyte close to metal, thus taking advantage of the signal enhancement.

The manuscript is clearly written, well-structured and it properly addresses most relevant literature already published on the subject.

Overall, data presented are technically sound. Materials and methods used have been described in sufficient detail to allow for others to reproduce the experiments reported.

As evinced by the scant literature on qualitative and quantitative SERS detection of analytes in serum and plasma, the chemical complexity of these biofluids represents a formidable challenge for SERS. The interaction between SERS-active nanostructured metallic surfaces and biofluids and the consequences of such interaction on SERS sensing of target molecules are still not completely understood. The authors seem to be well aware of these issues, and main claim of their manuscript is a new way of functionalizing SERS metal surfaces with a which makes possible to obtain spectra of analytes in complex biofluids.

*Although the same group published in 2015 a paper in which they present a simpler but similar approach to quantify fructose in presence of serum proteins (Sun et al. *ACS Nano*, 2015, 9 (3), pp 2668-2676), the results reported in this manuscript are definitely unprecedented: by using their "generalized platform" based on a hierarchical surface functionalization, the authors show that a range of structurally different molecules can be directly and indirectly detected (and quantified) in undiluted plasma. Everyone with experience in SERS knows that this detection, especially in undiluted plasma or serum, is far from trivial. In that sense, the general method presented in this manuscript represents a considerable step forward for the application of SERS to real-life samples. Thus this manuscript is likely to be of considerable importance for all the SERS community, and perhaps to further extend the interest for SERS to a broader public (e.g. clinicians).*

In spite of these unquestionable merits, however, this manuscript also presents some points of major and minor concern which in my opinion should be considered.

MAJOR REMARKS

The authors provide ample and convincing evidence to support the fact that structurally different molecules can be detected in undiluted plasma by using their approach. However, the limits of applicability of their approach are not mentioned. I wonder how much this research is really translational, and if it does really have the potential to be transferred into clinics, for real-life plasma samples.

In particular, I do have concerns about the following points.

*1. The concentrations detected for the different molecules are rather high, if compared with those expected for real-life applications. Spectra with recognizable doxorubicin (DOX) bands (in buffer solutions) are shown for concentrations down to a 0.1 μM (Fig. 3b), whereas plasma spiked with 20 μM DOX, yields a SERS spectrum (Fig. 3a) with well-recognizable bands. Assuming that the limit of quantification (which was not determined) will reasonably be higher than the last detectable concentration of the "free" drug, this method is likely not to be widely useful for Therapeutic Drug Monitoring (TDM) of DOX, since typical peak concentrations in plasma (of total DOX, i.e. HAS-bound + free) are between 6 and 50 ng/mL (approx. corresponding to 10 to 80 nM) of total drug (see Sottani et al. *J. Chromatogr. B* 2013, 915-916, 71-78; Ackland et al. *Clin. Pharmacol. Ther.* 1989, 45, 340-347), with some exceptional cases having a total DOX concentration up to 950 ng/mL (approx. 1 μM , see Piscitelli et al. *Clin. Pharmacol. Ther.* 1993, 53, 555-561). Same arguments hold true for the other drugs directly detected: amitriptyline (AH), carbamazepine (CARB) and phenytoin (PHEN). They were all detected at a 20 μM concentration. Fructose was indirectly detected at 5 mM, two order of magnitude more than the usual normal physiological concentration (30-50 μM , according to the human metabolome database, www.hmdb.ca).*

*Considering that in SERS the electromagnetic enhancement factor is rapidly decreasing with the distance from the metal surface (Camden et al. *Acc. Chem. Res.* 2008, 41, 1653-1661; Kennedy et al. *J. Phys. Chem. B* 1999, 103, 3640-3646), in the approach making use of self-assembled monolayers of thiols proposed by the authors, the ability of attracting small molecules in presence of complex biofluids while avoiding fouling, is gained at the expense of enhancement (i.e. distance from the metal), leading to a decrease of limit of detection, limit of quantification and sensitivity with respect to "bare" metal surfaces.*

I wonder if the sensitivity and limit of quantification of the method proposed can be good enough to be useful for a real TDM. Or, perhaps, if the method presented is an important step, but nevertheless just a step, toward a SERS-based TDM. The authors are kindly invited to critically discuss this point, perhaps adding a brief paragraph in their manuscript.

We appreciate these insightful comments. The normal target level of carbamazepine (CARB) is 5-12 mg/mL (about 20-50 μM); when we evaluated the limit of detection (LOD) for CARB, we obtained ~ 0.5 μM , which is far below this target level. Similarly, the normal target level of phenytoin (PHEN) is 10-20 mg/mL (about 40-80 μM). Our LOD found for PHEN is 1 μM , which is also lower than the clinically relevant range. Therefore, we believe our method is sufficiently sensitive for real TDM applications of these two drugs. For amitriptyline (AH),

the LOD of our system is 0.05 μM , which is at the upper limit of the normal range of AH (120-150 ng/mL, i.e., 0.04-0.05 μM). While not yet sufficient for TDM of AH, our method is presently capable of checking for AH toxicity or overdose. All these new results are discussed on page 6, line 4, and the SERS spectra are provided in Fig. S6 in the Supporting Information.

The normal blood level of DOX is around 10-80 nM, and it's true that our system cannot yet achieve this low limit of detection. DOX, being an anticancer drug, does not typically require therapeutic monitoring of its blood level and was primarily intended as a representative analyte in this work. That said, our system could still be useful for DOX monitoring at particularly high blood levels or around tumor tissues where DOX accumulates.

More broadly, the sensitivity and LOD presently achieved with our method is good for many drugs but not perfect for all of them. Even when functional SAMs are integrated, detection sensitivity depends highly on the intrinsic Raman activity of the drug as well as the enhancement of the SERS system. Our group is endeavoring to develop more sensitive SERS substrates as well as a long-range SERS system to overcome these limitations and expand the surface chemistry to more drugs of interest. In addition, we have studied the LOD of fructose using 4-MPBA in our previous work, and found it to be 20 μM (*Analytical Chemistry* 86.5 (2014): 2387-2394). We used 5 mM fructose in this work to better illustrate the normal functionality of the 4-MPBA probe after mixing within the pCBAA modification.

2. The novelty of the method presented consists in a two-fold role of the hierarchical SAM: i) preventing the fouling of the SERS-active surface and ii) directly attracting (or probing indirectly) the analytes of interest close to the same surface. The authors present SAMs incorporating different "attracting thiols" for the direct SERS detection of different molecules (i.e. alkane-based ATRP initiators for DOX, 3-mercaptopropionic acid for AH and CARB, 1-propanethiol for PHEN). What should be done in case another drug is to be detected, which is structurally different from the ones investigated in this manuscript? I wonder how general is this method: is there a rationale for selecting such "attracting" elements or it is a matter of trial and error? In the latter case, the method would lack general applicability. The authors are kindly invited to explicitly address this issue in the manuscript.

As we mentioned, we are trying to expand our system to more drugs, but our system is not perfect for all drugs. For example, we also tried to detect salicylic acid using positively charged "attracting" elements. However, the intrinsic Raman activity of salicylic acid is too low to be detected even with this modified sensor surface. Currently, our rationale is based on physical interactions (such as charge and hydrophobicity) and chemical/biological interactions.

3. Closely related to the previous point is the "selectivity" of the method proposed. For instance, can doxorubicinol (the main metabolite of doxorubicin, also present in the blood) be detected as well (and distinguished from doxorubicin) with the same method used for doxorubicin? Can the indirect method used to detect fructose be used to distinguish other sugars as well? The authors are invited, if not to present results from new experiments, at least to comment about this aspect.

Thanks for this suggestion. In the revised manuscript, we included SERS detection results of doxorubicinol (DOXol). The discussion was added on page 4, line 42. DOXol did show a different SERS spectrum than DOX, and DOX and DOXol can be distinguished in the mixture as shown in Fig. 3f and g. These results also demonstrated the multiplex detection capacity of our system.

MINOR REMARKS

Besides these major point of concern, there are few minor points which deserve some attention.

1. Band assignments for SERS spectrum of plasma (Table S1 in Supporting Information) are partially incorrect. Assignment of most bands in SERS spectra of plasma and serum is still debated, and is made mostly on the basis of the Raman shifts as found in literature from Raman spectra. On the other hand, some bands were previously assigned on the basis of direct comparison with the SERS spectra of the metabolites (Bonifacio et al. Anal. Bioanal. Chem. 2014, 406, 2355-1365, ref.15 of the manuscript), and can be relied on: bands at 631, 881 and 1138 cm⁻¹ are due to uric acid, and the band at 955 is due to hypoxanthine and/or adenine (whose SERS spectra can be very similar). Collagen is very unlikely to be found in plasma, so it should not be considered as a possible candidate for assignments. The authors are kindly invited to consider some changes in Table S1.

Thanks for the suggestion. We have corrected the assignments of the SERS spectrum of blood plasma according to the references. Please see the revised Table S1 in the Supporting Information.

2. The approach proposed involves complex, time-consuming, custom or expensive instrumentation or procedures (organic synthesis of initiators and of CBAA; EBL; UV-Ozone cleaner; custom glass tube reactor under nitrogen; deoxygenation steps; strong vacuum; etc.). In my opinion, the ease of substrate preparation and functionalization, the related costs and the substrate's shelf-life should be considered, and possibly addressed in the manuscript, when describing a new SERS method. The authors are kindly invited to address these issues, perhaps adding a sentence or a short paragraph in the manuscript.

We agree that all these factors have to be taken into account when develop a new SERS method. We have added a short discussion about the ease and cost of this system on page 7, line 27. We have also been developing simple surface functionalization approaches using “graft to” methods, in which, for example, zwitterionic polymers containing dopamine moieties are able to attach to many types of surfaces (metal and oxides) (Biosensors and Bioelectronics 25.3 (2010) 2276-2282).

3. Error bars are shown in Fig. 3c, but I could not find the numbers of independent measurements for each concentration. The authors are kindly invited to add this information to the manuscript.

The error bars represent the standard deviation of three replicates. We have added this information in the revised manuscript in the figure caption of Fig. 3.

In view of both merits and points of concern expressed above, my opinion is that, provided the authors adequately address all the remarks, an amended version of this manuscript should be definitely considered for publication in Nat. Comm.

Reviewer #3 (Remarks to the Author):

This manuscript describes the development of a surface chemistry protocol that allow the detection of small molecules from blood samples by SERS. The manuscript is interesting and the results are compelling. The originality, however, is not that high since there are several examples in the literature of SERS detection from whole blood using different strategies (Nanomedicine : nanotechnology, biology, and medicine Volume: 12 Issue: 3 Pages: 633-41; ANALYST Volume: 141 Issue: 7 Pages: 2165-2174; ELECTROPHORESIS Volume: 37 Issue: 5-6 Pages: 786-789, CHEMICAL SCIENCE Volume: 6 Issue: 7 Pages: 4247-4254 Published: 2015). The use of zwitterionic surfaces to decrease surface fouling is also not new (JOURNAL OF MEMBRANE SCIENCE Volume: 475 Pages: 469-479 ; ADVANCED MATERIALS INTERFACES Volume: 3 Issue: 6 Pages: 646-646). In any case, the level of the science presented here is good and the work might be considered for publication after the minor points below are considered:

1. The use of the 520 cm⁻¹ as an internal standard was a good idea. However, the band appears sometimes too weak in the spectra (see, for instance, figure 3). The concern is that division by a small (and noisy) number can lead to more errors. However, no errors are reported in the plots. Some statistical consideration should be added. It is actually not very clear that the 520 cm⁻¹ is constant from the data in figure 3.

The SERS spectra in Fig. 3b are subtracted data of DOX without the Si peak at 520 cm⁻¹. For better illustration, we have included the corresponding raw data with pCBAA peaks (including Si peak) in Fig. S3 in the revised Supporting Information. The Si peaks at 520 cm⁻¹ in all spectra are strong, which are good enough to be used as internal standards for reliable quantification. The results show small error bars based on this method as shown in Fig. 3c.

2. Although the experiments with single analytes were interesting, it is likely that blood samples from patients might contain mixtures of small molecules. How selective would be the layers for mixtures?

Our system is capable for multiplex detection. Two or more drugs (fit for same surface chemistry) can be detected simultaneously and can be distinguished by their own characteristic peaks. In the revised manuscript, we have included a multiplex detection experiment to differentiate doxorubicin (DOX) and its metabolite doxorubicinol (DOXol). The results are shown in Fig. 3f and g. More discussions have been added on page 4, line 42.

Reviewer #2 (Remarks to the Author):

In the revised version of the manuscript, the authors successfully addressed most of the issues raised by the Reviewers, in some cases by adding convincing results, after performing additional experiments.

Thus, in my opinion, the revised version of this manuscript is fit for publication in Nature Communications.

Reviewer #3 (Remarks to the Author):

The authors have addressed my previous concerns properly. I would then like to recommend the publication of this manuscript.

Point-by-Point Response

REVIEWERS' COMMENTS:

Reviewer #2 (Remarks to the Author):

In the revised version of the manuscript, the authors successfully addressed most of the issues raised by the Reviewers, in some cases by adding convincing results, after performing additional experiments. Thus, in my opinion, the revised version of this manuscript is fit for publication in Nature Communications.

Response:

We appreciate the reviewer's former comments and suggestion that have helped us to improve our manuscripts.

Reviewer #3 (Remarks to the Author):

The authors have addressed my previous concerns properly. I would then like to recommend the publication of this manuscript.

Response:

We appreciate the reviewer's former comments and suggestion that helped us to improve our manuscripts.